

# Healthy eating promoting in a Brazilian sports-oriented school: a pilot study

Fábia Albernaz Massarani, Marta Citelli, Daniela Silva Canella and Josely Correa Koury

Instituto de Nutrição, Universidade do Estado do Rio de Janeiro, Rio de Janeiro, RJ, Brasil

## ABSTRACT

**Background**. Adolescents, particularly athletes, have high exposure to ultra-processed foods, which could be harmful to their health and physical performance. School environments are capable of improving eating patterns. Our study is aimed at capturing changes in students' food consumption three years after they enrolled at an experimental school, considered a model of health promotion in Rio de Janeiro city. We also aimed to depict the promising nature of the healthy eating promotion program implemented in the school and share the learnings from its implementation.

**Methods**. Our pilot study was a follow-up on the implementation of a school garden, experimental kitchen activities, and health promotion classes. We evaluated 83 adolescent athletes' food consumption twice during the study: at its beginning (2013) and end (2016), by administering a food frequency questionnaire (FFQ) that inquired about the frequency of foods consumed in the past week. To evaluate how effectively the activities were established, integrated, and sustained in schools, the Garden Resources, Education, and Environment Nexus (GREEN) tool was used, and the school's adherence to the school garden program was classified as high (scored 47 points out of 57).

**Results**. In 2013, 89 adolescents (mean ± SD 11.9 ± 0.4 years, 54% male) participated in the study, of which 83 continued until 2016 (14.8 ± 0.5 years, 55% male). In 2013, the mean frequency of raw salad and fruits consumption was 1.4 (CI [1.0–1.9]) and 4.3 (CI [3.8–4.9]) days per week, respectively. Three years later, the frequency of raw salad and fruits consumption was 2.2 (CI [1.6–2.7]) and 5.0 (4.5–5.5), respectively. Considering that five meals were offered at school (five days/week), it may be possible to assume that the program raised awareness on the importance of healthy eating.

**Conclusion**. Our results suggest that such integrated healthy eating promotion programs may improve adolescent athletes' eating habits, by increasing the frequency of their consumption of unprocessed foods. This pilot study's results inspired us to implement an expanded project at the municipal level. Since 2018, teachers who participated in this program are working with Rio de Janeiro's Municipal Secretary of Education for Coordination of Curricular Projects. Some learnings from this pilot study on implementing the garden/experimental kitchen project in this school are being applied in 65 schools of the municipal network: joint activities must be fostered among students, teachers, and parents; healthy eating needs to be a respected value among adolescent athletes and become an example for parents and teachers.

Corresponding authors
Marta Citelli, martacitelli@gmail.com, citelli@uerj.br
Josely Correa Koury, jckoury@gmail.com

## INTRODUCTION

Sports-related food marketing promotes the consumption of energy-dense, nutrient-poor products as ultra-processed foods (*Powell, Harris & Fox, 2013*; *Bragg et al., 2018*). Ultra-processed foods are industrial formulations with high energy density, rich in fat, simple carbohydrates, and nutrients directly related to a higher incidence of chronic diseases, such as obesity (*Monteiro et al., 2016*; *Fardet, 2018*). Natural or minimally processed foods, on the other hand, are sources of micronutrients that are beneficial for health (*Louzada et al., 2015*; *Monteiro et al., 2016*). Adolescents, athletes in particular, are vulnerable to external factors such as marketing strategies for ultra-processed foods (*Bragg et al., 2018*). Adolescent athletes consume a high quantity of low-nutritious quality foods, particularly sugar sweetened drinks, and a low quantity of vegetables and water, leading to an insufficient intake of micronutrients and fiber, and an elevated quantity of refined carbohydrates (*Burrows et al., 2016*; *Sousa et al., 2008*).

To improve sportspersons' diet quality, effective strategies need to be identified. Schools are recognized as supportive environments to promote a healthy diet (*Briggs, Safaii & Beall, 2003*; *Scherr et al., 2017*; *Hoque et al., 2016*). School interventions should adopt an approach that integrates parents and the whole school with the curriculum, leading to hands-on experience (*Storz & Heymann, 2017*). School garden programs (*Ozer, 2007*) and experimental kitchens emerge as strategies to achieve these goals (*Robinson-O'Brien, Story & Heim, 2009*; *Wang et al., 2010*; *Scherr et al., 2017*). Despite the studies on school gardens, little is known about how gardens can be effectively integrated and maintained in a school (*Burt, Koch & Contento, 2017*). According to Ozer's definition (*Ozer, 2007*), a well-integrated school garden program includes three main components of implementation: a garden site and gardening activities, formal curriculum (including "hands-on" education), and involvement of parents and the community. To identify how to put school gardening components and the successful school garden integration into operation, the GREEN tool was developed to test the operational school gardening components proposed by Ozer (*Burt, Koch & Contento, 2017*). School garden programs could be considered as multi-component interventions to promote healthy eating in the school environment. Instead of evaluating the isolated effects of each component, it is crucial to consider the integrated effects of the actions that make up the intervention (*Scherr et al., 2017*; *Burt, Koch & Contento, 2017*).

Considering that adolescent athletes are vulnerable to the consumption of unhealthy foods, nutritional education programs, such as school garden programs, may impact food choice and eating habits (*Christian et al., 2014*; *Wells, Myers & Henderson, 2014*). However, in general, studies combining nutritional education actions with experimental gardens and kitchens are of short duration and are conducted without proper integration with the school curriculum (*Utter, Denny & Dyson, 2016*), which can reduce their effects. So, the aim of this study was to explore changes in students' food consumption three years after their enrollment at an experimental school considered as a model of health promotion in Rio de Janeiro city, and to present lessons learnt from the school's implementation of the promising healthy eating promotion program.

## MATERIALS & METHODS

### Study design and participants

This pilot study was developed in an experimental full-time sports-oriented public school located in the central region of Rio de Janeiro. The school's pedagogical model includes three axes: academic excellence, support for the student's life project and education values. Healthy eating promotion activities are inserted in the context of these three axes.

The students were enrolled at school in February 2013 and the present study began in August 2013. All students in the 6th grade ($n = 102$) were invited to participate in the study. However, only 89 students participated in the data collection (baseline) study in 2013, out of which 83 were followed up until 2016, when they were in the 9th grade. Data collection always occurred in the second semester (from August to December).

The students, unlike those of other Brazilian public schools, undertook 100 min of sports daily. The modalities offered were swimming, judo, badminton, athletics, soccer, volleyball, and table tennis. In addition to sports, they also attended physical education classes of 50 min per week, as in other Brazilian schools. The adolescents in this study were classified as adolescent athletes, and they were enrolled at a school with specific sports purposes. They participated in training, skill development, and were engaged in competition, according to the definition found in *Sports Dietitians Australia Position Statement: Sports Nutrition for the Adolescent Athletes* (*Desbrow et al., 2014*).

The school offered five meals a day: breakfast (bread/crackers/cookies, milk, coffee, margarine); morning snack (cookies, industrialized juice); lunch (rice, beans, a protein, cooked vegetables, fruit); afternoon snack (fruit or processed juice); late afternoon snack (milk or industrialized juice, fruit, bread/crackers/cookies/cake, a protein). The diet composition had a macronutrients energy distribution according to the *WHO/FAO (2003)* recommendation (carbohydrate 55–75%, protein 10–15%, fat 15–30%). This diet composition was not changed throughout the study period.

### Healthy eating promotion actions

Once a week, they had classes on Health Promotion (a mandatory subject) and were exposed to school gardening and experimental kitchen activities. "Health Promotion" aimed at raising awareness of the importance of cultivating healthy habits. Two elective subjects ("Gardening" and "Flavor and Art") were implemented in the school's curriculum with the objective of attracting students to participate in gardening and cooking activities. The school garden and experimental kitchen were built at the school for the promotion of scientific research with a grant from a Brazilian agency.

The GREEN tool (*Burt, Koch & Contento, 2017*) was used to assess the degree of school garden integration, and also as quality control on the actions implemented in the school. This integrated program's actions, description and categorization according to *Ozer (2007)* are summarized in Table 1.

The activities of the elective subjects were organized by the respective teachers of Arts ("Flavor and Art"), Mathematics or Physical Education ("Gardening"). The classes were supported by a group of researchers from Rio de Janeiro's Nutrition Institute of the State University. Every week, the art teacher selected some healthy preparations, often made with

Massarani et al. (2019), *PeerJ*, DOI 10.7717/peerj.7601

**Table 1  Integrated school program's actions, description and categorization.**

| Actions of the school multi component program | School garden domain categories (Ozer[19]) | Components of the school garden domains (Ozer[19]) related to the actions of the school multi component program | Description |
|---|---|---|---|
| Planning and construction of the school garden and the experimental kitchen | • Garden logistics<br>• School culture | • Planning and stablishing the physical space<br>• Characteristics of the physical space<br>• Garden care and upkeep<br>• Crop diversity<br>• Budget and funding<br>• Network and outside organization<br>• Administrative support<br>• Organization staff structure<br>• Volunteer and parent involvement<br>• Social events and activities<br>• Food environment and policies | The construction of the physical space of the school garden and the experimental kitchen was done with the direct involvement of teachers and parents through their participation in the planning, conception of ideas, choice and acquisition of materials. A group of parents, who were beneficiaries of a municipal fellowship program, helped to carry out the maintenance of the garden. Each semester, this group was partially renewed. |
| Development of materials for nutritional education | • Student experience<br>• School culture | • Connection with curriculum<br>• Time spent in the garden<br>• Additional learning opportunities<br>• Administrative support<br>• Organizational staff structure<br>• Food environment and policies | Three new disciplines were added to the school curriculum: "Health Promotion" (mandatory); "Gardening" (elective); "Flavor and Art" (elective). "Health Promotion" was taught once a week, for 50 min, during the school year. Elective disciplines were taught once a week, for 50 min. All students participated in the elective disciplines for one semester, at least. |
| Seeding, planting, maintenance, harvesting, food preparation and tasting by the students | • Garden logistics<br>• Student experience | • Connection with curriculum<br>• Activities<br>• Engagement<br>• Tasting opportunities<br>• Additional learning opportunities | Each week, students participated in activities that were proposed by the "Gardening" discipline, which involved sowing, planting, maintaining or harvesting the vegetables. The "Flavor and Art" discipline was held on a weekly basis and aimed to promote students getting in touch with healthy foods and learning new ways to create tasty culinary preparations. Harvests of the school garden were used by the "Flavor and Art" discipline. "Flavor and Art" was coordinated by the art professor. |
| Interdisciplinary classes | • Student experience | • Additional learning opportunities | Use of the school garden and experimental kitchen to teach other subjects. |

vegetables and spices harvested from the garden. Many times, the culinary preparations were chosen with the intention of diminishing rejection of some vegetables, such as eggplant (*Solanum melongena* L.) or bitter tomato (*Solanum aethiopicum* L.), both collected in the school garden. Students were also encouraged to suggest culinary preparations from their own homes.

### Food consumption

Food consumption was evaluated twice during the study: at the beginning (2013) and at the end (2016), by administering an FFQ that inquired about the frequency of foods consumed in the past week (from "never" to "everyday" in the previous seven days) of 12 food items (beans, cooked vegetables, raw salad, fruits, milk—that were natural or minimally processed foods; French fries, fried snacks, processed meat, crackers, cookies, candies, soft drinks—that were processed and ultra-processed foods). This questionnaire had been previously validated (*Tavares et al., 2014a*).

### Data analysis

Characteristics of students were described as frequency or mean and standard deviation (SD). Results from the GREEN tool were described as absolute values.

Food consumption was determined by considering the frequency of consumption per week (prevalence, mean and 95% confidence interval).

### Ethical aspects

This study was approved by the Ethics Committee of the Pedro Ernesto University Hospital (CEP/HUPE 1.020.909) and the Municipal Secretariat for Education (07/005.242/14). Parents and the student participants signed the informed consent form.

## RESULTS

In 2013, 89 adolescents (mean ± SD 11.9 ± 0.4 years; 54% male) participated in the study, out of which, 83 continued until 2016 (14.8 ± 0.5 years, 55% male). Six students (7%) left the school and hence were not evaluated in 2016.

The GREEN tool classified the sports-oriented school as a well-integrated school garden since out of a total of 57 points, it scored 47 (resources and support score = 12/15; physical garden score = 13/15; student experience score = 16/18; and school community score = 6/9 points).

In 2013, the mean frequency of consumption of raw salad and fruits was 1.4 (CI [1.0–1.9]) and 4.3 (CI [3.8–4.8]) days per week, respectively. Three years later, the frequency consumption of raw salad and fruits was 2.2 (CI [1.6–2.7]) and 5.0 (4.5–5.5) respectively (Table 2).

Furthermore, the consumption of some ultra-processed foods (French fries, fried snacks, candies, and soft drinks) did not seem to have increased during the three years of adolescence. On the other hand, when ultra-processed foods, such as crackers and cookies were served at school, the frequency of consumption seemed to increase.

Massarani et al. (2019), *PeerJ*, DOI 10.7717/peerj.7601

**Table 2  Food consumption based on markers of unprocessed and processed foods.**

| Food intake markers | Prevalence (%) of frequency consumption | | | | | | | | | | | |
|---|---|---|---|---|---|---|---|---|---|---|---|---|
| | Never | | 1 or 2× per week | | 3 or 4× per week | | 5 or 6× per week | | Everyday | | Mean (95% CI) | |
| | 2013 | 2016 | 2013 | 2016 | 2013 | 2016 | 2013 | 2016 | 2013 | 2016 | 2013 | 2016 |
| Beans | 0.0 | 1.2 | 2.2 | 3.6 | 2.2 | 9.6 | 18.9 | 13.8 | 76.7 | 78.0 | 6.5 (6.3; 6.7) | 6.5 (6.2; 6.7) |
| Raw salad | 44.4 | 32.4 | 33.3 | 39.6 | 14.4 | 7.2 | 2.2 | 4.8 | 5.6 | 15.6 | 1.4 (1.0; 1.9) | 2.2 (1.6; 2.7) |
| Cooked vegetables | 18.9 | 20.4 | 34.4 | 32.4 | 17.8 | 10.6 | 12.2 | 20.4 | 16.7 | 15.6 | 2.9 (2.4; 3.4) | 2.9 (2.4; 3.5) |
| Fruits | 8.9 | 3.6 | 21.1 | 19.2 | 17.8 | 6.0 | 14.4 | 26.4 | 37.8 | 44.4 | 4.3 (3.8; 4.9) | 5.0 (4.5; 5.5) |
| Milk | 26.7 | 19.2 | 7.9 | 10.8 | 14.4 | 9.6 | 5.5 | 13.2 | 44.4 | 46.8 | 4.0 (3.4; 4.7) | 4.4 (3.8; 5.1) |
| French fries | 66.7 | 69.6 | 30.0 | 22.8 | 2.2 | 4.8 | 0.0 | 0.0 | 1.1 | 0.0 | 0.5 (0.3; 0.7) | 0.6 (0.3; 0.8) |
| Fried snack | 46.0 | 46.8 | 51.1 | 46.8 | 0.0 | 4.8 | 1.1 | 1.2 | 1.1 | 0.0 | 0.8 (0.6; 1.0) | 0.8 (0.6; 1.0) |
| Processed meat | 21.1 | 21.6 | 43.3 | 49.2 | 25.6 | 14.4 | 3.3 | 2.4 | 6.7 | 12 | 2.0 (1.6; 2.4) | 2.1 (1.6; 2.6) |
| Crackers | 33.3 | 34.8 | 46.7 | 38.4 | 12.2 | 11.8 | 1.1 | 7.2 | 6.7 | 8.4 | 1.6 (1.2; 2.0) | 1.9 (1.4; 2.4) |
| Cookies | 33.3 | 26.4 | 43.3 | 37.4 | 10.0 | 15.6 | 4.4 | 8.4 | 8.9 | 12.0 | 1.8 (1.4; 2.3) | 2.3 (1.8; 2.8) |
| Candies | 15.6 | 16.8 | 32.2 | 31.2 | 21.1 | 25.2 | 5.5 | 3.6 | 25.6 | 21.6 | 3.2 (2.7; 3.8) | 3.1 (2.6; 3.7) |
| Soft drinks | 24.4 | 18.0 | 45.6 | 48.4 | 22.2 | 15.6 | 1.1 | 8.4 | 6.7 | 4.8 | 2.0 (1.6; 2.4) | 2.0 (1.6; 2.4) |

**Notes.**
Food frequency questionnaire (FFQ)—inquired about the frequency of the past week's food consumption (from "never" to "everyday" in the previous seven days).
CI, confidence interval.

## DISCUSSION

This study found positive results in adolescent athletes' frequency of consumption of natural or minimally processed foods three years after they were enrolled in an experimental school in which the multi-component educational program was implemented. This result is important since ultra-processed foods have been pointed out as being unhealthy, rich in energy and poor in protective micronutrients, antioxidants and fiber (*Monteiro et al., 2016*; *Fardet, 2018*) and adolescent athletes need an adequate dietary intake due to growth, health maintenance, and optimal athletic performance (*Croll et al., 2006*).

Since studies on school gardens and experimental kitchens use different methodologies and tools to evaluate their results, making direct comparisons is difficult (*Robinson-O'Brien, Story & Heim, 2009*; *Davis, Spaniol & Somerset, 2015*). Most authors rate as successful the increase in the consumption of fruits and vegetables by students who maintain direct contact with the cultivation, harvest, and preparation of the produce in the vegetable garden (*Burt, Koch & Contento, 2017*), as well as with the culinary preparations and tasting (*Lakkakula et al., 2010*; *Chen et al., 2014*).

Some factors of the present program's modus operandi seem to have been decisive for its acceptance and approval by students, teachers, and parents throughout the years in which it was executed. The involvement and communication among us and all those involved was the focus of this pilot study. All implemented actions, such as the choice of crop diversity or how the kitchen should be designed, were based on the suggestions given by parents, students, and teachers.

In general, studies combining nutritional education activities, experimental gardens and kitchens are of short duration and are conducted with elementary students (*Davis, Spaniol & Somerset, 2015*), without proper integration with the school curriculum (*Somerset et al., 2005*). One of the mechanisms deployed for students' adherence to the present program was the creation of elective subjects involving gardening and culinary activities. In addition, activities involving other disciplines such as Math, Arts and Biology were often carried out in an integrated way with the activities of this healthy eating program. To our knowledge, no study or long-term interventions have been performed with adolescent athletes enrolled in a sports-oriented school. Studies conducted in different countries with the introduction of an integrated garden in the school curriculum strengthen the actions of nutritional education (*Morris & Zidenberg-Cherr, 2002*; *McAleese & Rankin, 2007*) and resulted in an increased consumption of vegetables (*McAleese & Rankin, 2007*). The actions that focus on practical tasting or cooking activities, such as the use of experimental kitchens, have also proved to be effective in the preference and consumption of natural food by students in the United States (*Lakkakula et al., 2010*; *Chen et al., 2014*). By promoting interest in food preparation, this type of intervention stimulates students to make healthier food choices both at school, and at home with the family (*Hyland et al., 2006*; *Lakkakula et al., 2010*), and the preparation of vegetables and fruit juices starts getting more frequent (*Wang et al., 2010*; *Chen et al., 2014*). To our knowledge, none of these studies included culinary activities for adolescent students.
In the present study, changes in the frequency of consumption of natural or minimally processed foods corroborate the previous findings and confirm the relevance of the school multi-component actions on the overall quality of student nutrition. However, despite the changes in the frequency of consumption of natural foods, the frequency of consumption of soft drinks, French fries, fried snacks, and candies did not vary appreciably during the study. Nonetheless, the percentage of regular consumption of processed and ultra-processed foods was also high among adolescents in recent national surveys (*Tavares et al., 2014b*; *Borges et al., 2018*). *Aerenhouts et al. (2008)* found that consumption of soft drinks contributed considerably to higher energy intake in adolescents practicing field training. Soft drink consumption might negatively affect physical and sprint performance capacity (*Aerenhouts et al., 2008*). Male adolescents who consumed soft-drinks tended to have an unbalanced high-fat and low-carbohydrate diet. Female adolescents who consumed soft-drinks had a higher body-fat percentage than those who did not consume (*Sousa et al., 2008*). Despite knowing the harmful health effects of the consumption of soft drinks, food companies' use of sports to promote unhealthy consumption of food/beverage by young athletes is associated with healthy products (*Bragg et al., 2018*). This fact intensifies the need for implementation of public health policies, such as school garden programs. Furthermore, it is known that the consumption of soft drinks among adolescents is greater when their parents are habituated to consuming it at home (*Yee, Lwin & Ho, 2017*). Therefore, in our study, parents' participation in programs to promote healthy eating should be expanded beyond their participation in the care and organization of school gardens and the semiannual meetings. Creating a context of respect with multiple adults, in which adults know students' core values and are empathic about underlying causes of behavior was one of the lessons learned from this pilot study and has been considered as an important step to influence adolescent behavior (*Yeager, Dahl & Dweck, 2018*).

In contrast, when ultra-processed foods, such as crackers and cookies were served at school, the frequency of consumption seemed to increase. This result shows that, besides the actions carried out by this program, the menu offered by the municipal school feeding network should be based on "real" food because students acquire habits that are formed at school. It is especially important considering that 25% to 31% of the students were beneficiaries of the Bolsa Família program, which is a Brazilian cash transfer program.

Some limitations were observed in the present study. Food consumption data was obtained by administering a questionnaire that only elicited details on the frequency of consumption of food markers of a healthy diet (based on natural or minimally processed foods) or unhealthy diet (ultra-processed foods). This questionnaire has been used in Brazil to monitor the health of children and adolescents by the Brazilian Ministry of Health (2009, 2012 and 2015). Furthermore, it is a simplified FFQ focusing on food markers related to risk and prevention of chronic diseases, but not covering the diversity of the diet.

Additionally, considering that this was an experimental sports-oriented school, it was not possible to separate the effects of the healthy eating promotion program from the other actions that this type of school promotes. The lack of a comparison group was the major limitation of the study. Therefore, it will be important to carry out a controlled

study involving full-time sports-oriented schools where this program has not yet been implemented.

Nevertheless, the results of this pilot study inspired the implementation of an expanded project at the municipal level. Since 2018, teachers who participated in this program are working with Rio de Janeiro's Municipal Secretary of Education for the Coordination of Curricular Projects. Some lessons from this pilot study on implementation of this school's garden/experimental kitchen project are being applied in 65 schools of the municipal network: collaborative actions by students, teachers and parents; making healthy eating a respected value among adolescent athletes and setting an example for parents and teachers.

To sum up the strengths of this pilot study: it helped us to understand how to achieve improvements in dietary behaviors and sustain the garden-based programs in schools; our school multi-component program was formulated considering the school garden domains proposed by *Ozer (2007)*; integration between the school and the school garden program was tested using the GREEN tool (*Burt, Koch & Contento, 2017*); all participants were homogeneous regarding sports training in specific modalities offered by the school. Additionally, our study is in agreement with the Academy of Nutrition and Dietetics, School Nutrition Association, and Society for Nutrition Education and Behavior's position that recommends specific strategies for healthy food (*Academy of Nutrition and Dietetics, 2018*), as well as the International Olympic Committee consensus statement on youth athletic development that emphasizes dietary education for young athletes leading to optimal eating patterns to support health, normal growth and sport participation demands, with emphasis on a balanced diet (*Bergeron et al., 2015*).

## CONCLUSIONS

In conclusion, an adequate school environment, made up of facilities that encourage health promotion actions, structured subjects, trained teachers, sports orientation, and the development of an integrated curriculum, may help adolescent athletes to improve their eating habits. The contact of the adolescent athletes with the school garden and the experimental kitchen, as well as their involvement with local activities, may contribute to increasing their consumption of healthy foods (natural or minimally processed foods) and decrease their consumption of unhealthy foods (processed and ultra-processed foods). This is of extreme relevance at this stage of life, especially considering the nutritional demands generated by sports. Finally, improving the dietary pattern and quality of food consumption of these athletes will help them to promote health by optimizing performance and providing positive benefits beyond the adolescence phase.

## ACKNOWLEDGEMENTS

The authors would like to thank: Professors—Angelica Carvalho, Rosana Moraes and José Edmilson da Silva, Daniel Correa and Ana Christina Quintella who facilitated the implementation of healthy eating actions in school, all students who participated in this study, and Carolyne Rosado and Erica Pereira Leite for their assistance in data collection.

### Funding

This work was supported by Fundação Carlos Chagas Filho de Amparo à Pesquisa do Estado do Rio de Janeiro (No. E-26/112.637/2012; E-26/190.239/2013). The funders had no role in study design, data collection and analysis, decision to publish, or preparation of the manuscript.

### Grant Disclosures

The following grant information was disclosed by the authors:
Fundação Carlos Chagas Filho de Amparo à Pesquisa do Estado do Rio de Janeiro: E-26/112.637/2012, E-26/190.239/2013.

### Competing Interests

The authors declare there are no competing interests.

### Author Contributions

- Fábia Albernaz Massarani, Marta Citelli, Daniela Silva Canella and Josely Correa Koury conceived and designed the experiments, performed the experiments, analyzed the data, prepared figures and/or tables, authored or reviewed drafts of the paper, approved the final draft.

### Human Ethics

The following information was supplied relating to ethical approvals (i.e., approving body and any reference numbers):

This study was approved by the Ethics Committee of the Pedro Ernesto University Hospital and the Municipal Secretariat for Education (1.020.909, 07/005.242/14).

### Data Availability

Raw data are available as Supplemental Files.

### Supplemental Information

Supplemental information for this article can be found online at http://dx.doi.org/10.7717/peerj.7601#supplemental-information.

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
