# Peer review of "Healthy eating promoting in a Brazilian sports-oriented school: a pilot study"

_PeerJ, doi:10.7717/peerj.7601_

## Round 0.1 · original submission · Major Revisions

The reviewers have made some quite challenging comments and queries about your manuscript. I have added my own comments below. As an intervention study, the conclusions being drawn are not able to be justified given the lack of a comparison group (there are several alternative explanations for within-group changes, as discussed below) and the within-group changes do not strike me as entirely convincing (promising perhaps, but not convincing at this stage). See the comments from Reviewer #1 about the use of the phrase “an intervention study” as well.

I think the manuscript could be very usefully reworked as a pilot/feasibility study, discussing the programme in more detail (as requested by both Reviewer #1 and Reviewer #2), presenting lessons learned in implementing and evaluating the programme for this particular type of school, showing the changes in reported behaviours as sufficient motivation for moving to a controlled trial (with a cautious interpretation of these changes for the time being), and outlining your future research plans for this area. I think that this would be a significant undertaking but if you feel you can and would like to revise the current manuscript, please respond to each of the reviewers’ comments and mine below. I think the resulting manuscript could be a useful addition to the literature and I hope you take every advantage of the reviewers’ helpful comments.

Due to the lack of a comparison group, the interpretation of the results needs to be tempered considerably (see also comments from Reviewer #1). You cannot, for example, based on the data you present, rule out these changes being entirely or partially due to a) natural aging effects, b) effects of moving to higher competition levels in their chosen sport(s), c) other changes at the school level (this is a challenge with school-based research more generally), d) other changes at the local or national level, or e) Hawthorne effects. There are likely to be other possibilities that I haven’t listed here. Statements such as “Our results showed that this type of integrated food education program can contribute substantially to the improvement of adolescent athletes eating habits by increasing the frequency of consumption of healthy foods and decreasing the frequency of consumption of unhealthy foods” (Lines 41–44) read as causal and cannot be justified based on p-values without a comparison group, particularly given the clustered nature of the data (the same point applies to the discussion and conclusions). Any comment on the promising nature of the intervention in this context, and I do find the results to be sufficiently interesting and promising, should be based around observed effect sizes (and 95% CIs if analyses are used that provide these). Would these observed changes be plausible under a non-intervention counterfactual? This would be much easier to discuss if you use parametric approaches (such as mixed models or GEEs) and statistical power would be enhanced by using ordinal and continuous outcomes. I think the study would be better seen as a pilot/feasibility study, which would move the focus to the lessons learned during the programme implementation and the promising nature of the programme.

Given the 102 students in the 6th grade at the start of 2013, it would be useful to compare the baseline characteristics of those who declined participation (n=13) if possible, to those with follow-up data (n=83) and those without (n=6). A table of study participant characteristics is needed (age, sex/gender, any other salient information) and this could be used to show these subsets of the students.

What was the sample size calculation/basis? If this is a pilot/feasibility study, this could be based around saturation of experiences, problems, adverse events (e.g., rule of 3), etc. and obtaining sufficiently precise estimates for designing a larger trial.

How was multiplicity accounted for in the study (Line 142), or were comparisons only for 2013 to 2016, or was the multiple testing not accounted for? The additional analyses on “zero days” would then further increase the Type I error rate. Again, this would be problematic for a trial of the intervention, but less concerning in a pilot/feasibility study.

The timing of the FFQs is not clear (this point is also raised by Reviewer #1 and Reviewer #2). It would seem to be a very high burden to do this every week for three years, and if so, I’d want to see compliance data, or was this done once/a small number of times per phase? Note that it is extremely unlikely (and some statisticians would argue impossible) for FFQ data to be multivariate normally distributed and so polychoric correlations would be needed for the factor analysis. Were cross-loadings checked also (Lines 162–164, note that there is at least one of these in Table 2 despite the rotation)? How was the number of factors determined (parallel analysis would be the standard)? There is also no discussion here of how the factor scores were compared over time, which on Lines 193–201 seems to be descriptive only. Reviewer #2 asks similar questions and as they suggest, this could be simplified into operationally defined sets of foods (which would avoid the issue of polychoric correlations).

The lack of substantial changes in the dietary patterns (Lines 193–201) observed from 2013 to 2016 does not appear to support the claims that the programme was beneficial (I’m not sure what “positive results…in the dietary patterns” you mean on Lines 204–205). An alternative might have been to develop the factors based on 2013 data and then apply these factor weights to 2015 and 2016, with analyses through mixed models/GEEs/RM-ANOVAs (see Reviewer #2 again).

Given the special nature of the school, which as Reviewer #2 notes, provides a gap in the literature for this study, I think you need to be more careful about comparing your data to national data (Lines 237–239). The challenges of generalising your findings is also mentioned by Reviewer #1. If these differences with national data were also seen in 2013, pre-intervention, this would support the interpretation that there are differences between your participants and typical adolescents, rather than that the intervention might have had an effect.

The major limitation of your study (Lines 249–253) has to be the lack of a comparison group. I think you need to substantially temper your interpretation of the results given this very significant limitation and explain here how the results you have obtained (and I don’t mean just the statistical results which become much less important in this situation) can be used in future research.

As noted by both Reviewer #1 and Reviewer #2, there is some work needed on the writing and I’ll make some suggestions (an incomplete list) below. These appear to overlap with those suggested by Reviewer #2.

Line 58: “candied” Is this missing a word or should this be simply “candy” or “candies”?

Line 68: Do you mean something like “…integrates INTO the curriculum…” or “…BE integrated WITH the curriculum…” here? (Line 89 would be consistent with the second option.)

Line 90: Do you mean “minimize” (make as small as possible) or “reduce” here?

Line 91: I suggest deleting “the” (just before “adolescent’s” and changing “adolescent’s” to “adolescents’”.

Line 124: “Students who started the 6th grade (2013) had never had contact with…” needs rewriting (e.g. “…did not have contact…”) and do you mean “started the 6th grade DURING 2013)?

Lines 146–147: No “the” is needed before “principle components analysis”, but this would be “as THE extraction method”.

Line 174: Whenever presenting values, it should be immediately clear what these represent. While the reader might guess that these are mean ± SD, this should be made explicit if it is the case.

Line 184: Related to the above, it is often helpful for readers when the test that is the source of the p-value is included in the results (at least for the first instance).

Line 207: Just “fiber”.

Line 244: There seems to be an extra hyphen in “consuming-soft-drinks”.

Lines 245–248: This does not appear to be a complete sentence.

Line 247: “demonstrates” might work better than “intensifies” here?

Line 247: “need FOR” would be more usual than “need of”.

Lines 247–248: Do you mean “FOR EXAMPLE school garden programS”?

Line 250: Perhaps “…only allows US to measure THE frequency…”

Line 252: “focusing ON” would be more usual than “focusing in”.

Lines 254: I’m not sure what you mean here.

Figure 1: I’d add “a” to those foods without evidence for change so that the reader knows that there was no evidence as opposed to missing letters. While some software, such as Stata, treats blanks as distinct (and so different to other blanks), the usual treatment is to show at least one letter for every group.

Figure 2: I’m not sure why you have labels such as “a”, “ab”, and “ac” for crackers. Sharing an “a” would indicate no statistically significant difference between any of the three times. The same point applies to candies. You seem to have this correct for soft drinks though.

Table 1: “beneficiaries” rather than “beneficiary”.

Table 1: “administered” rather than “ministered”.

Table 1: I’m not sure what you mean by “elaborate” here, “create” or “enhance”?

Table 2: This shows “Factor loadings” not “Factorial load”.

Reviewer 1 ·

Basic reporting

Overall, this was a longitudinal study that focused on improvements of adolescent athletes' dietary habits based on the integration of a garden, kitchen, and elective courses in the school.
There were a few areas that I thought needed to be addressed:
Abstract:

Methods: suggest removing the “an interventional study” and keep only the longitudinal. The second sentence of the methods needs to be revised, suggest, “This was considered a multi-component study as participants were exposed to a garden….
Include how often you collected the FFQ information.
Include the age range of your participants

Results: if you use increased, please use ‘decreased’ instead of reduced. This sentence, ”In relation to the dietary patters, in 2013 and 2016 two similar patterns were identified: a healthy pattern, composed by natural or minimally processed foods and an unhealthy pattern, characterized by processed and ultra-processed food items” is not clear. Please reword.

Conclusion: use same terms, in the background and methods it was referred to as a multi-component intervention. Technically based on the various interventions you included, it was not food only. Maybe it was a component of the curriculum, but a nutrition / health program is a better term. Even though it was mentioned school is an ideal environment, there was an increase in soft drink consumption and only 2% of the 83 participants reduced their snack intake. Also, this was only for athletes, so the conclusion has to be more for that group.

Introduction:
In the first paragraph, the authors first indicated that ultra-processed foods are high in fats and carbs, but for the minimally processed have high amounts of micronutrients. Be consistent with either the focus being on macronutrients or micronutrients. The reviewer understands the issues with ultra-processed foods, but there needs to be more information about adolescents, especially concerning adolescents who are athletes as that is the argument for this study.

There is discussion about school gardens and a little bit about kitchen. Also, there is not much about if these gardens are traditionally placed in high schools or research has been predominately in elementary schools nor the stats behind the success of these, even if they are short term.

Line 60: if there are statistics to indicate how low the consumption of fruits and vegetables are, include that within this sentence.

I would suggest you have an individual, not associated with this manuscript to read for grammatical/technical errors as there were quite a few.

Experimental design

In the background of the abstract, For the aim, healthier/healthy, etc is an obscure word when talking about diets. Suggest nutritious, nutrient-dense, etc. Also the aim is wordy, suggest “The aim of this study was to determine athletic adolescent’s food consumption after implementation of a multi-component school intervention.”

Methods:
Overall, this section was sparse. Further development should be on the gardening of what was included/taught, and the kitchen of what students did. Also, expand on what these 2 elective courses were, who taught them and the expertise of the individual teaching all these interventions. There needs to be more information about if the students participated in these events each week for the 3 years or how the curriculum was set up. Essentially what you had in table 1 should have been concisely written in this part.

Even though the participants completed a weekly FFQ, did they do it consistently for 3 years, please clarify. Also, ultra-processed would also include sports bars, energy drinks, etc. However, there does not appear to be that information on this FFQ, please indicate why that was not included on the questionnaire.

Validity of the findings

Results:
Lines 196 and 197, identify how many participants fell into one of these 2 patterns.

Discussion:
It was mentioned there was a correlation between healthy eating and soft drinks, but in this study, soft drink consumption rose regardless of if students ate more cooked vegetables and raw salad. There was not much explanation as to why these interventions may have led to participants eating more nutritious products and why they continued to drink regular soft drinks.
The other limitation is the small number of participants. Even though this is a different type of school, the results cannot be generalized to other adolescents.

Reviewer 2 ·

Basic reporting

I suggest having a native English speaker proofread the manuscript to catch several spelling and grammatical errors. For example, Introduction, line 58, should be “Data from a Brazilian survey… more candies”; line 86; line 176
It is unclear what line 60-62 adds. I am not seeing the connection between the paragraph beginning on line 65 and the subsequent paragraph content. I think the authors could remedy this by making an explicit link between adolescents, school sports and food in schools. Line 84 is repetitive.
Line 87 – I question the relevance of this line of background given that the study did not examine the effect of school garden programs on athletic training and performance.
The structure of the article conforms to an acceptable format.

Figure 2: it may be preferable to include a comma between letters for the foods for which there was more than one statistical difference over timepoints indicated as “ab”, “ac” (so “a,c” “a,b”)
The submission represents an appropriate ‘unit of publication’.

Experimental design

Authors should be more specific about what they define as “healthier” (line 92). So the students attend a school that emphasizes more sports activities and dedicates more of the curriculum to physical education practice and study. The authors raised the issue of marketing unhealthy foods to athletes. Are they suggesting that these students are more amenable to marketing messages?
What really makes this study different then is the study population. Otherwise it is really similar to previous studies that it is testing the effect of a school-garden intervention on dietary consumption.
Suggest providing more of a definition/description of what a “sports-oriented” school is. Please also provide greater detail as to what the students received in the different intervention components. For example, what occurred in the experimental kitchen and what were the elective subjects that were implemented?
Line 89 – wouldn’t this study have been more effective and been better able to address your point about the utility of the length of the intervention and the integration within curriculum if it had been designed to include some students 1) who participated for different lengths of time and 2) who received all components of the intervention, some components of the intervention compared with students who didn’t receive any intervention (a control group)?

Suggest revising line 97; students began in the study in 6th grade and were followed to 9th grade. The way this statement is currently written it sounds as if students in grades 6-9 participated.
Suggest making it clearer that the 450 students who were enrolled in the study was the parent study and that the current paper focuses on longitudinal change in 6th grade participants. Suggest providing additional detail on reasons for why 13 (102-89) students didn’t participate (was it that they didn’t agree to participate?). Furthermore, what is the definition of “participation” in the study? Is this being used synonymously with completed baseline data collection since all students presumably received the intervention?
Line 129 suggest clarifying as follows “food frequency questionnaire that asked about past week consumption of 12 food items” or similar. This was obtained on three separate occasions. How were the food items included on the questionnaire selected? Did the authors measure amount/quantity in addition to frequency?
Line 134 suggest indicating that dietary patterns will be described below.
Line 144 please provide more information on how the absence of consumption (zero days) was investigated.
Line 146 Is it defensible to derive a dietary pattern from only 12 items? I’m not sure I would use “dietary pattern” to describe what was being derived from the factor analysis because a “dietary pattern” usually refers to the sum of what someone is consuming. Is it really necessary to use a factor analysis when the authors were comparing natural/minimally processed versus processed/ultra-processed foods and it is fairly straightforward which of the 12 food items would be categorized into each group.
Line 159 It is not clear if the factor analysis was used each time for the three time points to derive patterns or if the patterns derived using baseline data were then applied to subsequent time points? Please clarify.
Line 193 I am really not understanding what the authors did with the factor analyses. Please clarify the approach. So factors were created at each of the three time points and they didn’t differ from each other? What does this tell you? Wouldn’t it have made more sense for the authors to create factors at baseline and then look to see if the scores for participants changed over time? In other words, did their factor score for “healthy diet” increase with greater time in the intervention?

Validity of the findings

Line 189 – confusing. Unclear what the Please re-word.
The authors are overstating their findings. Really there were changes for increasing raw salad and cooked veg (but only in the third year) and decreasing fried snacks. Fewer students reported that they did not consume fruits over time. On the other hand, students consumed more soft drinks and processed meats. Results were mixed.
The authors should better discuss what the significance is of the study population.
Paragraph beginning on line 231 is repetitive. This paragraph includes results which should be moved to the Results section. This paragraph also includes statements requiring citations (line 242). Are the students in this school that emphasizes sports being marketed to?

Additional comments

The authors have not compellingly demonstrated what this study adds. If it is that a school-garden intervention is applied to a unique study population, then this should be further described. The methods are somewhat difficult to follow. This is a small study that collected limited data. The longitudinal nature is definitely a strength but the authors really don’t provide much detail on the intervention itself. The biggest problem I have with the manuscript is that the authors overstate their findings. The results are mixed at best and really don’t convincingly show a benefit of the intervention over time on dietary consumption.

---

## Round 0.2 · Minor Revisions

Well done with your revised manuscript. The reviewers have made only a few comments each. I don’t feel that these will require too much work on your part to address. I will ask you to response to these comments, along with those I list below in the revised version of the manuscript, which I would then hope to be able to accept.

Lines 28–29: Just a suggestion, but “…inquired about the frequency of foods consumed in the past week.” might be slightly easier to read. See also Lines 140–141.

Line 33: You could change the order here so the reader knows what the values represent just before they read them, e.g. “mean ± SD 11.9 ± 0.4 years”. For Line 34, you could rely on the previous explanation and use just “14.8 ± 0.5 years, 55% male)” if you wish. See also Lines 160 and 161.

Line 35: You use a hyphen for the first CI here and an em-dash for the second (and on Line 37).

Line 62: Either “…and an elevated quantity of carbohydrates…” or “…and elevated quantities of carbohydrates…”

Line 66: I’m not sure about the “hence” here as the first point (before the “hence”) does not seem to lead to the second point (after the “hence”). These could perhaps be two separate sentences?

Line 74: As you’re talking about “a well-integrated school garden program”, you don’t need “in school garden programs” here—you are talking about a hypothetical single well-integrated programme.

Line 87: I don’t think you need this comma (after “consumption”).

Line 102: I suggest “Data collection” (without the “s”).

Line 108: I’m not sure what “They were enrolled with sports specific context…” means. Do you mean they were enrolled in classes with sports-specific content?

Lines 108–109: Is this a single list item (i.e., doesn’t need a comma but rather an “and”): “participated in training and skill development”?

Line 115: You could delete “of time” from here.

Line 126: Just “…as quality control…” (no “a” needed).

Lines 129–130: Is this a list of three things (Arts, Mathematics and PE)? If so, you would want a comma before “Mathematics”.

Line 140: “an FFQ” (rather than “a”). It would be a Food Frequency Questionnaire if said in full, but for the letters an Ef Ef Cue.

Line 149: Perhaps “Results from the GREEN tool…”

Line 151: What does “…and data collection.” mean here?

Line 164: For readers not familiar with the GREEN tool, perhaps you could provide the maximum scores for each score here also, e.g. “resources and support score=12 out of 15”, etc. This should help readers appreciate the higher student experience score and the lower school community score.

Line 166: “was” not “were” as this is for “the mean frequency of consumption…” (singular). You already use “was” back on Line 35 in this case. Also Line 168 (see Line 37). You could also have worded this, here and in the abstract” as “the consumption frequencies for…were…”.

Line 170: Perhaps “…increased during the three years of adolescence.”

Line 179: “…as being unhealthy…” (adding “being”)

Line 199: Do you mean “…activities in other disciplines…” (“in” rather than “of”, or alternatively “involving”)?

Line 200: I think “were often carried” will help the reader to remember this is about the present study better than “have often been carried”.

Line 204: Can you make “…a reinforcement in the actions of nutritional education” clearer.

Line 218: Rather than “…almost did not vary throughout the evaluated years.”, do you mean “…did not vary appreciably during the study.”?

Line 220: No need for the comma here.

Line 224: Just “diet” rather than “diets” (the singular matches the earlier “an”).

Line 229: “programs.” (adding “s”).

Line 233: “…the semiannual meetings that were held each year…” suggests both two per year (semi-annual) and once per year (each year).

Line 233: No need for the comma here.

Line 241: Perhaps “formed” rather than “fomented”.

Line 246: Perhaps “minimally processed” rather than “poorly processed” (as you use on Lines 56, 142, 177, 214, and 284).

Line 247: “…monitor the health…” (adding “the”)

Line 248: “…by the Brazilian…” (adding “the”)

Lines 254–255: Perhaps “…will be important to carry out a controlled study involving full-time sports-oriented schools where this program has not yet been implemented.” (important, controlled, involving, schools)

Lines 255–257: Sorry, but I don’t quite understand what you mean here: “Furthermore, a pilot study may help to solve the study’s other limitation of sample size, since the same students had to be examined in two different years.” The present study can be seen as a pilot/feasibility study, so how would a pilot study overcome this study’s limitations around sample size?

Lines 265–266: While I agree that this is a good thing from the present study, I wouldn’t call it a strength (a reason to trust or believe in the results). Perhaps, the long-term follow-up mentioned in this point is a strength though?

Table 1: “stablishing” should be “establishing”. Also “added to the school curriculum” (adding “to”). “during the school year” (“the” rather than “a”). Do you mean ‘by the "Flavor and Art" discipline’ (adding “the”)? “by the Art Professor.” (adding “the”).

Data files: GEO2013PE.xlsx has 90 rows of data (91 including headings). Shouldn’t this be 89 rows of data as stated in the manuscript or should the manuscript refer to n=90 for 2013? If there should be 90 rows of data, the percentages for 2013 change a little and the CI for fruit would be 3.8–4.9. For Milk, I wonder if the upper CI limit for 2016 was mis-rounded as using a Wald interval I get 5.06, rounded to 5.1 not 5.0 as in Table 2. Also for Fried Snack I get an upper limit of 1.1 not 1.0. Is there any way of adding ID numbers for the two files so the individual students can be matched between the two years? It would also be useful to include age and sex in the 2013 file so the results involving these variables can also be replicated.

Reviewer 1 ·

Basic reporting

The authors have significantly improved this manuscript, but there were a few small areas that needed to be restructured:

Introduction:
Lines 58-61: Please integrate into the first paragraph as this paragraph reads out of place. Suggest not to start off with Furthermore, but instead, "Adolescent athletes consume a high quantity of low-nutritious quality foods..." Suggest to use 'refined carbohydrates' as adolescents who are athletes need carbohydrates, but the focus is on whole grain carbs.

Experimental design

Thank you for expanding on this section, for lines 112-113, even though it was good to include the types of foods offered per each meal and snack, what is the overall composition? Based on 2000 kcals, with 50% from carbohydrates, 20% protein, 30% fat? The reviewer is not familiar with Brazil's school breakfast/lunch program, so it would be good for other readers to know how different/similar the nutrient composition is compared to their respective country.

Validity of the findings

No comment

Additional comments

No comment

Reviewer 2 ·

Basic reporting

Improved. Authors have taken reviews into consideration and made satisfactory edits to revised manuscript. Minor additional comments.
Line 112 and elsewhere “industrialized juice”; I am not sure what this is
Line 191 – better to use the Results section to refer to tables; use the discussion section to summarize or discuss implications of the data contained in tables

Experimental design

Improved. Authors have taken reviews into consideration and made satisfactory edits to revised manuscript. No additional comments.

Validity of the findings

Improved. Authors have taken reviews into consideration and made satisfactory edits to revised manuscript. No additional comments.

---

## Author Rebuttal · Round 0.2

**Universidade do Estado do Rio de Janeiro**
**Centro Biomédico**
**Instituto de Nutrição**

Marta Citelli
State University of Rio de Janeiro
e-mail: martacitelli@gmail.com

May 8th, 2019

**MANUSCRIPT**
**(original title)**: School multi-component intervention improves healthy eating in adolescent athletes: a longitudinal study
**(current title):** Healthy eating promoting in a Brazilian sports-oriented school: A pilot study

**AUTHORS:** Fabia Albernaz Massarani, Marta Citelli, Daniela Silva Canella, Josely Correa Koury

To Dr. Andrew Gray - Editor of PeerJ,

We thank you for the opportunity to send this manuscript and also for your careful analysis and considerations. We also thank the reviewers for their valuable revisions. We are providing this cover letter to explain point-by-point the details of the revisions in the manuscript and our responses to your comments. The article has been reformatted as a pilot study. Following your suggestions, the focus has been moved to the particular features of this type of school, presenting the lessons learned in the implementation of the healthy eating program and its promising nature.

The teachers, who at that time were school principals, wrote a letter describing the political actions being developed in Rio de Janeiro as a consequence of this pilot study. I am not sure about the relevance, but it could perhaps be published as a supplemental material.

Thank you in advance for your attention.

Yours sincerely,

Dr. Marta Citelli

Editor comments (Andrew Gray)

MAJOR REVISIONS

The reviewers have made some quite challenging comments and queries about your manuscript. I have added my own comments below. As an intervention study, the conclusions being drawn are not able to be justified given the lack of a comparison group (there are several alternative explanations for within-group changes, as discussed below) and the within-group changes do not strike me as entirely convincing (promising perhaps, but not convincing at this stage). See the comments from Reviewer #1 about the use of the phrase "an intervention study" as well.

**Response:** We agree. Thank you for this suggestion. We change the presentation of the study transforming it into a pilot study. The term "intervention study" has been replaced by "pilot study" throughout the manuscript, including the title.

I think the manuscript could be very usefully reworked as a pilot/feasibility study, discussing the programme in more detail (as requested by both Reviewer #1 and Reviewer #2), presenting lessons learned in implementing and evaluating the programme for this particular type of school, showing the changes in reported behaviours as sufficient motivation for moving to a controlled trial (with a cautious interpretation of these changes for the time being), and outlining your future researchplans for this area. I think that this would be a significant undertaking but if you feel you can and would like to revise the current manuscript, please respond to each of the reviewers' comments and mine below. I think the resulting manuscript could be a useful addition to the literature and I hope you take every advantage of the reviewers' helpful comments.

Due to the lack of a comparison group, the interpretation of the results needs to be tempered considerably (see also comments from Reviewer #1). You cannot, for example, based on the data you present, rule out these changes being entirely or partially due to a) natural aging effects, b) effects of moving to higher competition levels in their chosen sport(s), c) other changes at the school level (this is a challenge with school-based research more generally), d) other changes at the local or national level, or e) Hawthorne effects. There are likely to be other possibilities that I haven't listed here. Statements such as "Our results showed that this type of integrated food education program can contribute substantially to the improvement of adolescent athletes eating habits by increasing the frequency of consumption of healthy foods and decreasing the frequency of consumption of unhealthy foods" (Lines 41–44) read as causal and cannot be justified based on p-values without a comparison group, particularly given the clustered nature of the data (the same point applies to the discussion and conclusions). Any comment on the promising nature of the intervention in this context, and I do find the results to be sufficiently interesting and promising, should be based around observed effect sizes (and 95% CIs if analyses are used that provide these). Would these observed changes be plausible under a non-intervention counterfactual? This would be much easier to discuss if you use parametric approaches (such as mixed models or GEEs) and statistical power would be enhanced by using ordinal and continuous outcomes. I think the study would be better seen as a pilot/feasibility study, which would move the focus to the lessons learned during the programme implementation and the promising nature of the programme.

**Response:** Thank you for these suggestions that greatly improved the manuscript. As mentioned above, it has been reformatted as a pilot study. We have discussed the program in more detail (lines 93-97, 110-123, 128-

135), presented lessons learned in implementing the program (lines 188-193, 196-200, 228-242, 264-265), showed the changes in reported behaviours as sufficient motivation for moving to a controlled trial (lines 250-256), and outlined our future research plans for this area (lines 257-263). Considering the reviewers suggestions and aiming to avoid analyzes with low statistical power, the focus has been moved to the particular features of this type of school, presenting the lessons learned in the implementation of the healthy eating program and its promising nature. Thus, the results were presented as prevalence (%), mean and confidence interval (CI) of frequency consumption (new table 2). Analysis for identification of the dietary patterns were excluded.

Given the 102 students in the 6th grade at the start of 2013, it would be useful to compare the baseline characteristics of those who declined participation (n=13) if possible, to those with follow-up data (n=83) and those without (n=6). A table of study participant characteristics is needed (age, sex/gender, any other salient information) and this could be used to show these subsets of the students.

**Response:** We don´t have the baseline characteristics of those 13 who did not participate. They were not present when data were collected. Those without follow-up data (n=6) abandoned the school and, because of this, were not evaluated in 2016. We have included this information in the "results" section (lines 159-161), where age and sex are also related.

What was the sample size calculation/basis? If this is a pilot/feasibility study, this could be based around saturation of experiences, problems, adverse events (e.g., rule of 3), etc. and obtaining sufficiently precise estimates for designing a larger trial.

**Response:** Sample size was based on the number of students entering the school in 2013 in 6th grade (n=102). This information was clarified in the "methods" section (lines 98-101).

How was multiplicity accounted for in the study (Line 142), or were comparisons only for 2013 to 2016, or was the multiple testing not accounted for? The additional analyses on "zero days" would then further increase the Type I error rate. Again, this would be problematic for a trial of the intervention, but less concerning in a pilot/feasibility study.

**Response:** Comparisons were only for 2013 and 2016. Given the small number of students in 2015 data collection, we discarded it. We agree with these opinions and the study has been transformed into a pilot study.

The timing of the FFQs is not clear (this point is also raised by Reviewer #1 and Reviewer #2). It would seem to be a very high burden to do this every week for three years, and if so, I'd want to see compliance data, or was this done once/a small number of times per phase? Note that it is extremely unlikely (and some statisticians would argue impossible) for FFQ data to be multivariate normally distributed and so polychoric correlations would be needed for the factor analysis. Were cross-loadings checked also (Lines 162–164, note that there is at least one of these in Table 2 despite the rotation)? How was the number of factors determined (parallel analysis would be the standard)? There is also no discussion here of how the factor scores were compared over time, which on Lines 193–201 seems to be descriptive only. Reviewer #2 asks similar questions and as they suggest, this could be simplified into operationally defined sets of foods (which would avoid the issue of polychoric correlations).

The lack of substantial changes in the dietary patterns (Lines 193–201) observed from 2013 to 2016 does not appear to support the claims that the programme was beneficial (I'm not sure what "positive results…in the dietary patterns" you mean on Lines 204–205). An alternative might have been to develop the factors based on 2013 data and then apply these factor weights to 2015 and 2016, with analyses through mixed models/GEEs/RM-ANOVAs (see Reviewer #2 again).

**Response:** Analysis for identification of the dietary patterns were excluded. We agree with these opinions and the study has been transformed into a pilot study. Furthermore, we tried to explain better about the FFQ collection (lines 137-144).

Given the special nature of the school, which as Reviewer #2 notes, provides a gap in the literature for this study, I think you need to be more careful about comparing your data to national data (Lines 237–239). The challenges of generalising your findings is also mentioned by Reviewer #1. If these differences with national data were also seen in 2013, pre-intervention, this would support the interpretation that there are differences between your participants and typical adolescents, rather than that the intervention might have had an effect.

**Response:** We have excluded comparisons with other studies. We have just presented it (lines 165-171).

The major limitation of your study (Lines 249–253) has to be the lack of a comparison group. I think you need to substantially temper your interpretation of the results given this very significant limitation and explain here how the results you have obtained (and I don't mean just the statistical results which become much less important in this situation) can be used in future research.

**Response:** We agree with you and have highlighted it in the discussion section (lines 152-156).

*As noted by both Reviewer #1 and Reviewer #2, there is some work needed on the writing and I'll make some suggestions (an incomplete list) below. These appear to overlap with those suggested by Reviewer #2.*

**Response:** Thank you. We have submitted the manuscript into an English writing revision. So, the following observations (italic) were corrected in the text:

*Line 58: "candied" Is this missing a word or should this be simply "candy" or "candies"?*

*Line 68: Do you mean something like "...integrates INTO the curriculum..." or "...BE integrated WITH the curriculum..." here? (Line 89 would be consistent with the second option.)*

*Line 90: Do you mean "minimize" (make as small as possible) or "reduce" here?*

*Line 91: I suggest deleting "the" (just before "adolescent's" and changing "adolescent's" to "adolescents'".*

*Lines 146–147: No "the" is needed before "principle components analysis", but this would be "as THE extraction method"*

*Line 124: "Students who started the 6th grade (2013) had never had contact with..." needs rewriting (e.g. "...did not have contact...") and do you mean "started the 6th grade DURING 2013)?*

*Line 207: Just "fiber".*

*Line 244: There seems to be an extra hyphen in "consuming-soft-drinks".*

*Lines 245–248: This does not appear to be a complete sentence.*

*Line 247: "demonstrates" might work better than "intensifies" here?*

*Line 247: "need FOR" would be more usual than "need of".*

*Lines 247–248: Do you mean "FOR EXAMPLE school garden programS"?*

*Line 250: Perhaps "…only allows US to measure THE frequency…"*

*Line 252: "focusing ON" would be more usual than "focusing in".*

*Lines 254: I'm not sure what you mean here.*

Line 174: Whenever presenting values, it should be immediately clear what these represent. While the reader might guess that these are mean ± SD, this should be made explicit if it is the case.

**Response:** We have included mean ± SD (lines 159-160).

Line 184: Related to the above, it is often helpful for readers when the test that is the source of the p-value is included in the results (at least for the first instance).

**Response:** The results have been presented as prevalence (%), mean and confidence interval (CI) of frequency consumption (new table 2). So, figures 1 and 2 were excluded.

Figure 1: I'd add "a" to those foods without evidence for change so that the reader knows that there was no evidence as opposed to missing letters. While some software, such as Stata, treats blanks as distinct (and so different to other blanks), the usual treatment is to show at least one letter for every group.

Figure 2: I'm not sure why you have labels such as "a", "ab", and "ac" for crackers. Sharing an "a" would indicate no statistically significant difference between any of the three times. The same point applies to candies. You seem to have this correct for soft drinks though.

**Response:** The results have been presented as prevalence (%), mean and confidence interval (CI) of frequency consumption (new table 2). So, figures 1, 2 and table 2 were excluded.

Table 1: "beneficiaries" rather than "beneficiary".

**Response:** We have fixed it.

Table 1: "administered" rather than "ministered".

**Response:** We have fixed it.

Table 1: I'm not sure what you mean by "elaborate" here, "create" or "enhance"?

**Response:** We have replaced with "create".

Table 2: This shows "Factor loadings" not "Factorial load".

**Response:** The results have been presented as prevalence (%), mean and confidence interval (CI) of frequency consumption (new table 2). So, figures 1, 2 and table 2 were excluded.

[# PeerJ Staff Note: PeerJ provides language editing services for a fee. If you would like a quotation for this service, please email us at editorial.support@peerj.com #]

**Response:** Thank you. We have submitted the manuscript into an English writing revision.

**Reviewer 1 (Anonymous)**

Basic reporting

Overall, this was a longitudinal study that focused on improvements of adolescent athletes' dietary habits based on the integration of a garden, kitchen, and elective courses in the school. There were a few areas that I thought needed to be addressed:

Abstract:

Methods: suggest removing the "an interventional study" and keep only the longitudinal.

**Response:** This study has been transformed into a pilot study.

The second sentence of the methods needs to be revised, suggest, "This was considered a multi-component study as participants were exposed to a garden….

**Response:** We have modified it.

Include how often you collected the FFQ information.

**Response:** We have included this information (lines 25-27).

Include the age range of your participants

**Response:** We have included this information (lines 31-32).

Results: if you use increased, please use 'decreased' instead of reduced. This sentence, "In relation to the dietary patterns, in 2013 and 2016 two similar patterns were identified: a healthy pattern, composed by natural or minimally processed foods and an unhealthy pattern, characterized by processed and ultra-processed food items" is not clear. Please reword.

**Response:** We have excluded dietary patterns.

Conclusion: use same terms, in the background and methods it was referred to as a multi-component intervention. Technically based on the various interventions you included, it was not food only. Maybe it was a component of the curriculum, but a nutrition / health program is a better term. Even though it was mentioned school is an ideal environment, there was an increase in soft drink consumption and only 2% of the 83 participants reduced their snack intake. Also, this was only for athletes, so the conclusion has to be more for that group.

**Response:** Thank you. We have excluded the term "multi component intervention". Instead, we have adopted "healthy eating promotion program" while considering that many actions developed by the school belong to this program and aimed to promote healthy eating. The population of the study, adolescent athletes, was highlighted (lines 59-62, 80).

Introduction:

In the first paragraph, the authors first indicated that ultra-processed foods are high in fats and carbs, but for the minimally processed have high amounts of micronutrients. Be consistent with either the focus being on macronutrients or micronutrients. The reviewer understands the issues with ultra-processed foods, but there needs to be more information about adolescents, especially concerning adolescents who are athletes as that is

the argument for this study.

**Response:** We have expanded this issue in the introduction (lines 59-62).

There is discussion about school gardens and a little bit about kitchen. Also, there is not much about if these gardens are traditionally placed in high schools or research has been predominately in elementary schools nor the stats behind the success of these, even if they are short term.

**Response:** Thank you. Because of your comment, we have highlighted that most studies have carried out nutritional education activities with children, for a short period of time (line 195). We did not find any study including culinary activities with adolescent students (lines 211-212).

Line 60: if there are statistics to indicate how low the consumption of fruits and vegetables are, include that within this sentence.

**Response:** The results have been presented as prevalence (%), mean and confidence interval (CI) of frequency consumption (new table 2). So, figures 1, 2 and table 2 were excluded.

I would suggest you have an individual, not associated with this manuscript to read for grammatical/technical errors as there were quite a few.

**Response:** Thank you. We have submitted the manuscript into an English writing revision.

Experimental design

In the background of the abstract, For the aim, healthier/healthy, etc is an obscure word when talking about diets. Suggest nutritious, nutrient-dense, etc. Also the aim is wordy, suggest "The aim of this study was to determine athletic adolescent's food consumption after implementation of a multi-component school intervention."

**Response:** This text has been rewritten (lines 85-88).

Methods:

Overall, this section was sparse. Further development should be on the gardening of what was included/taught, and the kitchen of what students did.

Also, expand on what these 2 elective courses were, who taught them and the expertise of the individual teaching all these interventions. There needs to be more information about if the students participated in these events each week for the 3 years or how the curriculum was set up. Essentially what you had in table 1 should have been concisely written in this part.

**Response:** We have expanded our explanation (lines 116-123, 128-135).

Even though the participants completed a weekly FFQ, did they do it consistently for 3 years, please clarify.

**Response:** We clarified it in the "Methods" section (lines 137-144).

Also, ultra-processed would also include sports bars, energy drinks, etc. However, there does not appear to be that information on this FFQ, please indicate why that was not included on the questionnaire.

**Response:** We used a previously validated questionnaire that was adopted by the Brazilian Institute of

Geography and Statistics. We have included this reference in the manuscript.

Validity of the findings

Results:

Lines 196 and 197, identify how many participants fell into one of these 2 patterns.

**Response:** Now, the results are presented as prevalence (%), mean and confidence interval (CI) of frequency consumption (new table 2). So, figures 1, 2 and table 2 were excluded.

Discussion:

It was mentioned there was a correlation between healthy eating and soft drinks, but in this study, soft drink consumption rose regardless of if students ate more cooked vegetables and raw salad. There was not much explanation as to why these interventions may have led to participants eating more nutritious products and why they continued to drink regular soft drinks.

**Response:** We also could not identify any factor that might have increased the intake of soft drinks. Any explanation would be purely speculative. Even so, the discussion has been increased (lines 215-219, 229-236).

The other limitation is the small number of participants. Even though this is a different type of school, the results cannot be generalized to other adolescents.

**Response:** We have corrected it throughout the manuscript.
* * *
Reviewer 2 (Anonymous)

Basic reporting

I suggest having a native English speaker proofread the manuscript to catch several spelling and grammatical errors. For example, Introduction, line 58, should be "Data from a Brazilian survey… more candies"; line 86; line 176. It is unclear what line 60-62 adds.

**Response:** Thank you. We have submitted the manuscript into an English writing revision.

I am not seeing the connection between the paragraph beginning on line 65 and the subsequent paragraph content. I think the authors could remedy this by making an explicit link between adolescents, school sports and food in schools. Line 84 is repetitive.

**Response:** We have modified the text (lines 59-62; line 80).

Line 87 – I question the relevance of this line of background given that the study did not examine the effect of school garden programs on athletic training and performance.

**Response:** We agree and have excluded it.

The structure of the article conforms to an acceptable format.

Figure 2: it may be preferable to include a comma between letters for the foods for which there was more than one statistical difference over timepoints indicated as "ab", "ac" (so "a,c" "a,b")

The submission represents an appropriate 'unit of publication'.

**Response:** Now, the results are presented as prevalence (%), mean and confidence interval (CI) of frequency consumption (new table 2). So, figures 1, 2 and table 2 were excluded.

Experimental design

Authors should be more specific about what they define as "healthier" (line 92).

**Response:** We have modified this text (lines 137-144).

So the students attend a school that emphasizes more sports activities and dedicates more of the curriculum to physical education practice and study. The authors raised the issue of marketing unhealthy foods to athletes. Are they suggesting that these students are more amenable to marketing messages?

**Response:** No, we are just saying that they are also amenable to marketing messages.

What really makes this study different then is the study population. Otherwise it is really similar to previous studies that it is testing the effect of a school-garden intervention on dietary consumption. Suggest providing more of a definition/description of what a "sports-oriented" school is. Please also provide greater detail as to what the students received in the different intervention components. For example, what occurred in the experimental kitchen and what were the elective subjects that were implemented?

**Response:** Thanks for these valuable suggestions. We have inserted some paragraphs to better explain it (lines 94-97, 116-123, 128-135).

Line 89 – wouldn't this study have been more effective and been better able to address your point about the utility of the length of the intervention and the integration within curriculum if it had been designed to include some students 1) who participated for different lengths of time and 2) who received all components of the intervention, some components of the intervention compared with students who didn't receive any intervention (a control group)?

**Response:** Yes, you're right. Unfortunately, we had many organizational problems that made designing the study difficult.

Suggest revising line 97; students began in the study in 6th grade and were followed to 9th grade. The way this statement is currently written it sounds as if students in grades 6-9 participated. Suggest making it clearer that the 450 students who were enrolled in the study was the parent study and that the current paper focuses on longitudinal change in 6th grade participants.

**Response:** It has been rewritten (lines 98-101).

Suggest providing additional detail on reasons for why 13 (102-89) students didn't participate (was it that they didn't agree to participate?). Furthermore, what is the definition of "participation" in the study? Is this being used synonymously with completed baseline data collection since all students presumably received the intervention?

**Response:** It has been rewritten (lines 98-101). Six students (7%) abandoned the school and, because of this, were not evaluated in 2016 (lines 160-161).

Line 129 suggest clarifying as follows "food frequency questionnaire that asked about past week consumption of 12 food items" or similar.

**Response:** Thank you for your suggestion (lines 138-140).

This was obtained on three separate occasions. How were the food items included on the questionnaire selected? Did the authors measure amount/quantity in addition to frequency?

**Response:** We used a previously validated questionnaire that was adopted by the Brazilian Institute of Geography and Statistics, evaluating only the frequency of consumption. We have included this reference in the manuscript (Tavares et al., 2014) (lines 143-144).

Line 134 suggest indicating that dietary patterns will be described below.

Line 144 please provide more information on how the absence of consumption (zero days) was investigated.

Line 146 Is it defensible to derive a dietary pattern from only 12 items? I'm not sure I would use "dietary pattern" to describe what was being derived from the factor analysis because a "dietary pattern" usually refers to the sum of what someone is consuming. Is it really necessary to use a factor analysis when the authors were comparing natural/minimally processed versus processed/ultra-processed foods and it is fairly straightforward which of the 12 food items would be categorized into each group.

Line 159 It is not clear if the factor analysis was used each time for the three time points to derive patterns or if the patterns derived using baseline data were then applied to subsequent time points? Please clarify.

Line 193 I am really not understanding what the authors did with the factor analyses. Please clarify the approach. So factors were created at each of the three time points and they didn't differ from each other? What does this tell you? Wouldn't it have made more sense for the authors to create factors at baseline and then look to see if the scores for participants changed over time? In other words, did their factor score for "healthy diet" increase with greater time in the intervention?

Validity of the findings

Line 189 – confusing. Unclear what the Please re-word.

The authors are overstating their findings. Really there were changes for increasing raw salad and cooked veg (but only in the third year) and decreasing fried snacks. Fewer students reported that they did not consume fruits over time. On the other hand, students consumed more soft drinks and processed meats. Results were mixed.

The authors should better discuss what the significance is of the study population.

**Response:** Considering the reviewers and editor suggestions and aiming to avoid analyzes with low statistical power, the focus has been moved to the particular features of this type of school, presenting the lessons learned in the implementation of the healthy eating program and its promising nature. Thus, the results were presented as prevalence (%), mean and confidence interval (CI) of frequency consumption (new table 2). Analysis for identification of the dietary patterns were excluded.

Paragraph beginning on line 231 is repetitive. This paragraph includes results which should be moved to the

Results section. This paragraph also includes statements requiring citations (line 242). Are the students in this school that emphasizes sports being marketed to?

**Response:** We fixed it (lines 215-219). We have included missing citations. Students are not being marketed.

Comments for the Author

The authors have not compellingly demonstrated what this study adds. If it is that a school-garden intervention is applied to a unique study population, then this should be further described. The methods are somewhat difficult to follow. This is a small study that collected limited data. The longitudinal nature is definitely a strength but the authors really don't provide much detail on the intervention itself. The biggest problem I have with the manuscript is that the authors overstate their findings. The results are mixed at best and really don't convincingly show a benefit of the intervention over time on dietary consumption.

**Response:** As mentioned above, the study has been reformatted as a pilot study. We have discussed the program in more detail (lines 93-97, 110-123, 128-135), presented lessons learned in implementing the program (lines 188-193, 196-200, 228-242, 264-265), showed the changes in reported behaviours as sufficient motivation for moving to a controlled trial (lines 250-256), and outlined our future research plans for this area (lines 257-263).

---

## Round 0.3 · accepted · Accept

Thank you for your constructive revisions, which have addressed all comments relating to the previous version of the manuscript. I am delighted to accept this version of your manuscript. I’ll note a minor typo in Table 1 (“Planning and Construction of the school garden and the experimental kitchen” has a spurious capitalisation of “Construction”) which you can correct during the proofing stage. I’ll also note that URLs in your references are a mixture of active links and ordinary text, but again this can be sorted in the proofing of your manuscript.

I look forward to hearing more from your group about further development and implementation of this promising intervention.

Reviewer 1 ·

Basic reporting

The authors have addressed all the minor revisions identified by the reviewer. Overall, this is a great manuscript and study.

Experimental design

No comments

Validity of the findings

No comments

Additional comments

No comments